# The Effect of Front-of-Pack Nutritional Labels and Back-of-Pack Tables on Dietary Quality

**DOI:** 10.3390/nu12061704

**Published:** 2020-06-06

**Authors:** Helene Normann Rønnow

**Affiliations:** Department of Food and Resource Economics, University of Copenhagen, Rolighedsvej 23, 1958 Frederiksberg C, Denmark; hnr@ifro.ku.dk

**Keywords:** home-scan data, front-of-pack label, nutritional tables, food labels, dietary quality, difference-in-difference, panel data

## Abstract

A healthy diet is important to prevent lifestyle diseases. Food labels have been proposed as a policy tool to improve the healthiness of food choices, as they provide information about nutritional content and health attributes which may otherwise have been unknown to the consumer. This study investigates the effect of food labels with different formats on dietary quality by using home-scan panel data and difference-in-difference methods to compare the change in dietary quality over time for households that start to use food labels with households that do not use labels. I find that the use of front-of-pack (FOP) nutritional labels increases overall dietary quality, which is driven by reduced intake of added sugar and increased intake of fiber. The use of back-of-pack (BOP) nutritional tables does not influence dietary quality. There is no additional benefit to overall dietary quality by using both labels. However, the results indicate that there could be a benefit of using both labels on certain nutrients. The results imply that additional policies are needed to supplement food labels in order to improve dietary quality.

## 1. Introduction

A healthy diet is important to maintaining good health throughout all life stages. An unhealthy diet can lead to obesity, underweight, and diet-related diseases such as diabetes, cardiovascular diseases and cancer [1]. Therefore, it is important that policy makers guide consumers towards making healthy food choices, since an unhealthy diet is expensive for society in terms of medical costs and loss of labor and productivity [2,3,4,5]. One policy tool that can help to achieve healthier diets is food labels. The main purpose of food labels is to inform consumers about the nutrients contained in products and their overall healthiness, which are otherwise unobserved characteristics [6,7,8]. Obtaining knowledge about the nutritional content of each product will help consumers to plan their diet according to their preferences and health goals. Food labelling offers several advantages compared to other policy instruments. For instance, food labels do not restrict consumer choices and they reduce information search costs, since consumers can obtain the nutritional information while shopping [9]. Food labels are placed on food products and the format of the labels varies from back-of-pack (BOP) numerical nutrition tables and ingredient lists to graphical and colored front-of-pack (FOP) symbols [10,11]. This paper investigates the effect of using FOP labels to the effect of using BOP nutrition tables on dietary quality as well as the effect of using both types of labels.

BOP labels mostly consist of nutrition tables or nutrition facts as well as ingredient lists. The nutrition table shows the content of carbohydrates, protein and fat and potentially other nutrients per 100 gr or serving. BOP tables are located on the back of the product and are mandatory in many countries, e.g., the US and EU [12]. Several studies have examined whether the use of nutritional tables is associated with higher dietary quality by using non-experimental survey data: Guthrie et al. [13], Kim et al. [14] and Variyam [15] find that the use of nutrition tables or parts of the nutrition table influences the intake of some nutrients, e.g., fiber, iron and cholesterol, while Kim et al. [16] find that the use of nutrition table information is associated with a 4–6 point increase in a 100 point Healthy Eating Index. The above-mentioned studies are based on cross-sectional data, where food intake is calculated from dietary recalls. Patterson et al. [17] use several cross-sectional datasets to look at the effect of nutrition tables. They find that the implementation of the Nutrition Labelling and Education Act (NLEA) only has a small effect on snacking behavior, but not on other dietary components (e.g., milk fat, cereal calories, cereal fiber, etc.).

FOP labels are meant to supplement the BOP nutrition table as a summary of the nutritional content as FOP labels summarizes the information in the BOP table. The nutritional information can be summarized either per nutrient or as an overall evaluation of the healthiness of the product [10]. FOP labels vary in format from Guideline Daily Amount (GDA) to Traffic Lights (TL) and logos, e.g., the Keyhole, Choices, and Smart Choice labels [18,19]. A large body of experimental literature has compared the effect of FOP labels in different formats according to their ability to facilitate healthy food choices [20,21,22,23,24,25,26,27,28,29,30]. Fewer studies have examined FOP labelling on the healthiness of food purchases by using actual food purchases: Smed et al. [31] investigate the effect of the introduction of a FOP label in Holland (Choices) using home-scan data and find an increase in consumers’ purchased share of labelled products within multiple product categories. The effect is evident in product groups with a high variation in healthiness. A study of the implementation of FOP labels on ready-to-eat cereals in the US shows that the labels increase the probability of choosing healthier cereal products [19,32]. Sacks et al. [33] find no effect on healthiness of sandwich and salad purchases after an introduction of a FOP label in a UK retailer. Similar findings are found by Boztug et al. [34].

Some studies have compared the effectiveness of FOP labels to nutrition tables. Watson et al. [30] use an online experiment and find that providing FOP labelling increases the probability of choosing a healthy alternative compared to nutrition tables only. They furthermore find that providing a FOP label reduces the propensity to use the nutrition tables. Neal et al. [35] provide experimental evidence from real food purchases by providing labelling on a smart-phone app. Comparing different FOP labels to nutrition tables, they find that warnings and recommendations increased the healthiness of food purchases. Similar setup and findings are found by Ni Mhurchu et al. [36]. Furthermore, studies have shown that nutritional labels and tables are more effective when consumers have a health goal in mind [29,37,38], and that consumers who use these labels are more health-oriented and have higher nutritional knowledge [39,40,41,42].

In general, most of the studies in the literature are based on experiments, while fewer are based on actual purchase or consumption data, to evaluate food label systems in use [11,18,43]. Based on a unique dataset with repeated questionnaires concerning stated label use issued to a panel of consumers combined with observed food purchases, this paper examines whether the use of FOP nutritional labels or BOP tables influences dietary quality by using a difference-in-difference design. The use of home-scan data and following the same household over a longer time horizon (~3 years) are novel in the literature on food labels, providing the possibility to use panel data methods and control for unobserved fixed effects to investigate a causal effect of label use on dietary quality. First, the effect of using FOP nutritional labels or BOP nutrition tables separately is investigated. In experiments, the use of nutrition tables and the use of nutrition labels are often treated as mutually exclusive treatments. However, in real life, they are not. From the data, it is shown that some consumers utilize both labels available when they shop for food. In an expanded analysis, it is investigated whether using both nutritional tables and labels have additional benefits to dietary quality.

I find that the use of FOP nutritional labels increases the overall dietary quality. However, the effect is small (0.2–0.3 in a Healthy Eating Index ranging from 0 to 24.5). This increase in overall dietary quality can be explained by a decrease in the intake of added sugar and an increase in fiber intake. There is no effect of using BOP nutrition tables on dietary quality, and no additional effect of using both labels on overall dietary quality. However, using both labels seems to affect the intake of some nutrients, e.g., a decrease in the intake of added sugar.

This paper is structured as follows: Section 2 describes the data and outlines the method and Section 3 presents the results. Lastly, Section 4 discusses the results and Section 5 summarizes the conclusions.

## 2. Materials and Methods

### 2.1. Data

In Denmark, several types of food labels are in use, but for this study, the following two types have been selected: Description of Content and nutritional labels. (The data for the current analysis contains information on the use of three label groups: Description of Content, health claims and nutritional labels. Health claims are excluded in this analysis due to a small share of label users (15 to 17% of households) and because Danish consumers are skeptical of the health claims [44]. Furthermore, health claims differ from the other labels because they do not necessarily help consumers to distinguish healthy alternatives within a product group, which is feasible with the Description of Content or the two selected nutritional labels.) The Description of Content (DoC) is an extensive BOP nutrition table, which is similar in format to the American Nutritional Facts (see Figure 1). The label displays the content of carbohydrates, fat and protein in the product as well as the ingredients it contains. This label is mandatory for all prepackaged food products.

Nutritional labels (NL) consist of graphical FOP labels, which indicate whether the product is a healthy option within its product group. In Denmark, two FOP nutritional labels are in use: the Keyhole (Nøglehulsmærket) and the Whole Grain logo (Fuldkornsmærket).

The Keyhole label is used in the Scandinavian countries and Lithuania [11]. In Denmark, the label is managed by the Danish Veterinary and Food Administration. The Keyhole label is used to identify healthy alternatives within a given product group and can be printed on pre-packed food products as well as fish, seafood, fruit, vegetables, cheese, bread, and unprocessed meat. The label is voluntary for the producers to print. A product is eligible for the Keyhole label when it complies with product group-specific criteria with respect to fat, sugar, salt and dietary fibers [45].

The Whole Grain label is managed by the Danish Whole Grain Partnership, which aims to increase the intake of whole grain products in Denmark. Members of the partnership can print the logo on their product if the product complies with product-specific standards on whole grain. Members include supermarket chains as well as producers of flour, bread, etc. The Whole Grain logo is printed on grain products only (rye bread, pasta, etc.) [46].

Both of the FOP nutritional labels provide an indication that the product is a healthy alternative within its product group, but do not provide a nutrition score or summary statistics of nutrients within the product. That is, they are evaluate labels indicating healthiness [11]. In that sense, they are similar to the Smart Choice and Great for You labels used in the US [19].

To investigate the effect of food labels on dietary quality, home-scan data from GfK Panel Services Scandinavia (hereafter abbreviated to GfK) was combined with two additional questionnaires and nutritional data from the National Food Institute (Technical University of Denmark) [47]. The GfK data consists of grocery shopping data, where a consumer panel of approximately 2500 distinct households regularly report their grocery shopping. The members are recruited when they respond to market surveys from GfK and GfK aims to sustain a representative panel (see comparison for the subset of GfK panel used in this analysis versus general population in Table A11). The panel members are incentivized to report their grocery shopping data with points, which they can use in a GfK-operated store. The selected data covers the years 2009–2016. The scanner data contains detailed information about the household’s purchases down to the brand level, and even the EAN level for some products, which makes it possible to merge it with nutritional data to evaluate the dietary quality of each purchase. The nutritional data contains information on calories, macronutrients (protein, fat, carbohydrates) and added sugar per 100 g, etc. The nutritional content is then multiplied by the volume of the purchased products to calculate the total consumption of each nutrient. The purchases are made at the household level, which means that it is not possible to access the consumption of individual household members. It is also not possible to observe how much of the purchased food is consumed or wasted. Therefore, I use consumption and purchase interchangeably in the remainder of this paper. The home-scan data provides the advantage that it covers all supermarkets compared to scanner data from a single supermarket chain. The home-scan data covers only food items bought in supermarkets and not food consumed at cafeterias or restaurants. However, since the selected food labels are only provided on packaged items in supermarket, the data is adequate for the purpose. The participating households annually report their background information such as age and gender of main shopper, household size, occupation, but also attitudes to cooking and shopping. These background variables are used as controls in the analysis.

Furthermore, two almost identical questionnaires were sent to the panel at the end of 2013 and 2015. The questionnaires contained questions about health habits, health attitudes, and health behavior including label use. In 2013, 1879 households answered the questionnaire, while 1774 households answered in 2015, which results in a balanced panel of 1210 households who answered both questionnaires (of the 1210 households, not all of them answered both questions on label use, which is why the total number of households differs when each label is evaluated separately (Table 1) and jointly (Table 2)). The use of food labels was measured by asking the respondents about their use of each of the two food labels during their most recent shopping trip. The questions were as follows:
“Did you read the Description of Content label on the products you bought?”
“Did you search for products with nutritional labels? (e.g., the Keyhole or Whole Grain label)”

For the first question, the response options were “Yes, for all items”, “For almost all items”, “For some items”, “No” and “Do not remember”. If the respondents answered the first three options, they are defined as users of the DoC. For the latter question, the respondents could answer either “Not at all”, “A bit”, “Some”, “A lot” or “Do not remember”. If the respondents answered “A bit”, “Some” or “A lot”, they are defined as users of NL. Respondents who answered “Do not remember” are coded as missing.

It is assumed that the use of food labels during the most recent shopping trip can be used to generalize whether a household uses food labels. I assume this for the following four reasons: first, the most recent shopping trip is easy for respondents to recall and, thus, they are less likely to avoid misstating use. Second, it is unlikely that households who rarely or never use labels will report label use during their most recent shopping trip. Therefore, they will be correctly identified as non-users when applying the most recent shopping trip. Another type of non-users consist of those who obtained the information in the past and, therefore, do not search for the information again. These can also be treated as non-users, since they do not search for new information in order to update their nutritional knowledge. Third, households who sometimes or often use labels are likely to search for food labels during their most recent shopping trip. If the household indicates that they used food labels during their most recent shopping trip, we know that they obtained new information in the observed period. Lastly, only 11% of the sample reported that their most recent shopping trip was unusual, which suggests that the most recent shopping trip is a good proxy for general shopping behavior. (The respondents are asked to describe their latest shopping trip using the statement “The shopping trip was as it usually is”. They can either answer “Agree”, “Partly agree”, “Neither agree or disagree”, “Partly disagree”, “disagree”. Their shopping trip is defined as unusual if they disagree or partly disagree with the statement. In that sense, it is not clear how it is different from their usual shopping trip.)

Approximately 25–27% of the households searched for products with NL, while 30–32% read the DoC. This is slightly higher than the share of users found in European countries by using in store observations (16.8%) [38], but lower than the share of users in the US using self-reported general use (75.8%) [14]. Because households are free to decide whether they use labels or not, some households use one label, while others choose to use both NL and DoC. Additionally, since there are two years where the households report their label use behavior, their behavior can be categorized as in Table 1 and Table 2.

Table 1 (left side) shows the number of households grouped according to their NL use status in 2013 and 2015. The use is abbreviated, where U stands for user of the label and N for non-user. For example, the UU refers to households that report using the given label in both years, while NU refers to households that do not report label use in the first year, but report label use in the second. The largest group consists of households that do not use labels in both years (*n =* 676). The second largest group consists of households that use the labels in both years (*n =* 158), while fewer switch behavior from use to non-use or vice versa. Table 1 (right side) shows the number of households grouped according to their use of the DoC in 2013 and 2015. Most households do not use the DoC in both years (*n =* 626), while fewer use DoC in both years (*n =* 210). Table 2 shows the label use status in each year where the use of each label is combined. As seen from Table 2, the largest group consists of households that do not use either of the labels in both years (*n =* 471). The second largest group is households that use both labels in both years (*n =* 96).

These different groups facilitate an examination of whether the dietary quality of users differs from that of non-users before label use by comparing the dietary quality of switchers to that of non-users. Furthermore, by comparing the change in dietary quality of those that switch behavior, it is possible to examine whether the use of food labels improves dietary quality. The methodology is described in detail in Section 3.

#### Dietary Quality

To measure dietary quality, I use a Healthy Eating Index (HEI) developed by Smed [48]. The Healthy Eating Index is calculated based on the household’s compliance with six of the official Danish dietary guidelines:Eat 6 pieces (600 g) of fruit and vegetables a day (potatoes are not included as a vegetable)—Children should eat 400 g.Eat fish several times a week (at least 200 g a week).Eat potatoes, rice, pasta or brown bread everyday (at least 3 g of fiber per 1 mega joule (MJ) of food consumed).Reduce consumption of sugar—especially from soft drinks, sweets and cakes (a maximum of 10% of total energy intake should come from added sugar).Reduce the amount of fats—especially from dairy products and meat (a maximum of 30% of total energy intake should come from fat).A maximum of 10% of total energy intake should come from saturated fat.

A score is calculated based on compliance with each of these targets. The HEI ranges from 0 to 24.5, where an index of 24.5 is assigned for compliance with all the dietary guidelines. For each dietary guideline, a score between 0 and 10 is assigned depending on how well the household complies with the guideline, where 0 is non-compliance and 10 is full compliance. The index is calculated as:(1)$$\mathrm{HEI}=\sqrt{\overset{6}{\underset{i=1}{\sum }}{\left({\mathrm{score}}_{i}\right)}^{2}}$$

The index is calculated on a monthly basis and then averaged to yearly observations due to seasonality. Since the HEI is composed of several components that do not have food labels (e.g., vegetables and the amount of fish eaten), I also examine the following three nutrients that are relevant for both labels: total fat, added sugar and fiber. To compare households of different sizes, the energy share for each nutrient is calculated as:(2)Energy share=((kJ per gram·gram of nutrient consumed)/Total kJ consumed)·100
where kJ per gram for fat is 37, kJ per gram for added sugar is 17, and kJ per gram for fiber is 8.

### 2.2. Method

To establish a causal relationship between label use and healthy food choices, the panel dimension of the data is utilized by applying a difference-in-difference setup [49]. Several issues need to be addressed to estimate the effect of food label use on dietary quality in the case where observational data is used rather than experimental data.

First, the use of food labels differs over time. The questionnaires allow us to examine the household’s use of the two different food labels at two points in time: end of 2013 and end of 2015. In this period, there was no change in labelling policy in Denmark and, thus, some households report label use in both years, others report no use in both years, and some households change their label use behavior over time. This is unlike the standard difference-in-difference setting in which no unit is enrolled in the program in the first time period, while some units are enrolled in the second period. This variation in label use over time allows the possibility to report label use in the first year, but not in the second—in other words, “to opt out” of label use.

Second, it is possible to use one or several labels at the same time. For example, some consumers choose only to look at the FOP labels, others look for elaborate information on the back of the product, and a third group utilizes all the available information and looks at both labels.

Third, the uptake of food label use is not randomly assigned. Therefore, the households decide whether to use labels or not themselves. This can cause selection bias issues if self-selected households differ in unobserved characteristics, e.g., preference for healthy food and nutritional knowledge compared to households which do not use food labels [13]. These differences in unobserved characteristics can influence both the propensity to use labels and dietary quality. For example, consumers with more knowledge about the link between diet and health will be more likely to search for ways to plan a better diet (use food labels) and eat healthy (dietary quality).

Fourth, there is a potential difference in start date of label use. Since there was no policy change in the observed period and there is a two-year time gap between the questionnaires, the households can either start to use food labels in the beginning of 2014 or in the end of 2015. Thus, it is unknown when during the two-year period the individual household started to use food labels.

In sum, the following four issues impose a threat to identification: defining clear treatment and control groups, multiple label use, selection, and difference in start date.

The applied identification strategy tackles the issues in the following way:

First, the change in label use status is used to identify the effect of label use in a difference-in-difference design, where a treatment group is constructed as the household that states that they do not use labels in the first year and that they use labels in the second year. I refer to these as “NL switchers” for those that start to use NL and “DoC switchers” for those that start to use the DoC. When the use of both labels is examined, I refer to the households that start to use both labels as “Both switchers”. In the baseline case, the control group consists of those households that do not report label use in both years. The households that report label use in the first year, but do not report it in the second year (the “opt-outs”), are a special case that needs attention. These households have obtained the label information previously and no longer actively search for information. The household still possesses this knowledge even though they do not continue to read food labels on a regular basis [40]. As a sensitivity check, this group is included in an extended control group along with the households that report label use in both years.

Second, the use of multiple labels is handled by including multiple treatments that are either use of one label or both labels. First, I examine the use of each label separately, not controlling for the other one, and then I combine the use of the two labels to see whether using both labels has an additional effect. In other words, households that start to use NL, DoC or both are evaluated as three separate programs and compared to a reference group that consists of non-users. The non-users of both labels are used as a control group in the baseline regression and as a robustness check, the households which exhibit constant use of one of the labels or both are included in an extended control group as a sensitivity check.

Third, selection is handled by including controls for observables for which there is selection. Furthermore, a treatment group fixed effect allows one to control for factors that are time invariant and unobserved and potentially correlated with propensity to use labels, such as preferences. As a sensitivity check, I add individual fixed effects to the regression instead of a group fixed effects. However, it should be noted that I cannot control for unobservable factors that are time variant unless they are correlated with observables, which we must assume is the case. For example, controlling for health attitude or diet status over time will be correlated with other attitudes and lifestyle changes. Since I cannot control for all variables that influence selection into label use, the results should be interpreted as average treatment effect on the treated rather than an average treatment effect.

Fourth, to handle the problem of an unknown switching time, dietary quality is evaluated post the last questionnaire, i.e., we compare dietary quality in 2013 and 2016. It is assumed that the effect of food labels on dietary quality is constant over time such that the effect does not differ for those households that switch at an early point compared to those who switched behavior at a later point.

The expected outcome can be written as:(3)$$E\left[Y_{t}^{d}|D=d\right]$$
where $Y_{t}$ is the outcome (dietary quality) in period t; d is the treatment status, which takes the value 1 in the case of treatment and the value 0 if not treated; D is treatment, which can take four values: NL,DoC, B, 0, which are NL, DoC, Both and no-use, respectively, depending on the regression.

To identify the effect of using food labels, I assume common trends. In the regression framework where NL and DoC are examined, this implies:(4)$$E\left[Y_{1}^{0}|D=DoC\right]-E\left[Y_{0}^{0}|D=DoC\right]=E\left[Y_{1}^{0}|D=0\right]-E\left[Y_{0}^{0}|D=0\right]$$
(5)$$E\left[Y_{1}^{0}|D=NL\right]-E\left[Y_{0}^{0}|D=NL\right]=E\left[Y_{1}^{0}|D=0\right]-E\left[Y_{0}^{0}|D=0\right]$$

Without treatment—in this case, label use—the switcher groups would have followed the same trend as the given control group. In the regression, where the use of both labels is included, the two above assumptions are maintained as well as the following assumption (In the regression where use of both labels is combined, NL switchers are defined as those who start to use NL in the observed period, while either not using DoC or using DoC in both years to keep the other label constant. Likewise, DoC switchers are defined as those who start to use DoC in the observed period, while either not using NL or using NL in both years.):(6)$$E\left[Y_{1}^{0}|D=B\right]-E\left[Y_{0}^{0}|D=B\right]=E\left[Y_{1}^{0}|D=0\right]-E\left[Y_{0}^{0}|D=0\right]$$

Whether these assumptions are reasonable is tested by using pre-label use periods (2009–2012) and by examining the trend for each of the label groups compared to the non-user group, graphically, and through regression analysis to check the assumption of common trends. Since we cannot test the assumption of parallel trend in the treatment period (2013–2016), the period before is used to test whether the assumption is reasonable [49].

To test for parallel trend in the pre-treatment period, I use the following approach from Autor [50]:(7)$$y_{it}=\beta_{0}+\beta_{1}\cdot year_{t}+\beta_{2}\cdot label_{i}+\beta_{3}\cdot year_{i}\cdot label_{t}+\beta_{4}x_{it}+u_{it}$$
where year is year dummies; label is treatment dummy/(dummies), indicating whether the household is a switcher of the label; year·label is the interaction between year and label dummy/(dummies). The regression (5) is run by OLS using the years 2009–2016, where 2013 is the reference year. The significance of each coefficient in the pre-treatment year is then examined to check significance $\left(\beta_{3}\right)$. The standard errors is clustered at the household level.

The difference-in-difference estimation regression is written as:(8)$$y_{it}=\theta_{0}+\theta_{1}\cdot year_{t}+\theta_{2}\cdot label_{i}+\theta_{3}\cdot year_{t}\cdot label_{i}+\theta_{4}x_{it}+u_{it}$$
where year is a time dummy, which is 1 in 2016; label is a dummy indicating whether the household is switcher of a particular label (NL, DoC, or both); year·label is the interaction between the treatment dummy and year; x is household characteristics; u is the error term. In this case, $\theta_{3}$ is the coefficient of interest, since it will show the treatment effect of food labels. Equation (8) is run by OLS using the years 2013 and 2016, where standard errors are clustered at the household level. In a robustness check, household fixed effects are included in Equation (8).

The included controls are household characteristics, which are assumed to influence dietary quality, but also controls that are assumed to be correlated with selection into label use. Previous research has established that attitudes and special diet requirements are important for label use [41,51]. Therefore, dummies for health attitude, special diet and an exercise variable are included in the list of controls. Furthermore, since demographic and socio-economic variables have also been found to influence the decision of whether to use food labels as well as dietary quality, they are added as a robustness check [16]. For an overview of the included controls, see Table 3. Stata 15.1 is used for data analysis.

## 3. Results

### 3.1. Summary Statistics

Table 4 shows the summary statistics for the switchers and the non-users in 2013. What is evident from the tables is that the switchers and non-users differ in several ways: the NL and DoC switcher groups contain a larger share of main-shoppers who care about healthy food. The NL switcher group contains a larger share of females, has a significantly different geographical composition, and exercises more than the non-users, whereas DoC switchers have a significantly smaller household size. The table shows that the two groups differ slightly on observables and, therefore, it is important to add these as controls in the regression. The summary statistics for all subgroups are shown in Appendix A
Table A1, Table A2 and Table A3.

### 3.2. The Effect of Food Labels on Dietary Quality

#### 3.2.1. Nutritional Labels

Figure 2 shows a graphical examination of common trends in dietary quality in the period 2009–2016. The first graph shows the change in the average HEI in the observed period. The NL switcher group and the non-users both experience a decrease in the HEI from 2009 to 2013. The average HEI remains constant for the NL switcher group from 2013 and onwards, while it decreases for the non-users. It should be noted that the switcher group has a higher HEI compared to the non-user group in the years before they start using labels and this is maintained after 2013. The regression test for common trends is included in the Appendix A
Table A4. None of the year and treatment interactions are significant before 2013 and, hence, we cannot reject common trends before 2013. The energy share from fat is constant for the NL switcher group in the observed period, whereas it is constant for the non-user group from 2009 to 2012 and increases from 2012 to 2013, but remains constant during the remainder of the period. The regression analysis of common trends shows that 2010 and 2012 are significant at the 5% and 10% level, respectively. However, since the year 2010 seems to be a single peak year, I maintain the assumption of a common trend. The energy share from added sugar is constant for both groups and shows a small downwards kink in 2016 for the NL switcher group. The regression results indicate that we cannot reject common trends from 2009 to 2012. Lastly, the energy share from fiber is constant for the non-user group, but increases slightly for the NL switcher group from 2013 onwards. The regression analysis of common trends shows that we cannot reject common trends in the energy share from fiber from 2009 to 2012.

Adding the households that use NL in both years and those households that use labels in the first year, but not the second year, to the extended control group still makes a valid control group with respect to the common trend assumption (see Appendix A
Figure A1 and Table A5).

Table 5 shows the difference-in-difference regressions for NL. Model (1) is the baseline model, where a dummy for the switcher group (NL), a time dummy (year) and the interaction (NL*year) are included. Model (2) and (3) add health and socio-economic controls to the regression. Model (4) uses individual fixed effects and model (5) shows model (3) with the extended control group. The first part of the table examines the overall dietary quality as assessed by the HEI. The coefficient on the group dummy indicates that switchers initially have higher dietary quality than the non-users, which is robust across models. However, the difference decreases from 0.4305 to 0.1997 when socio-economic and health controls are added. This aligns with the fact that switchers have a positive health attitude compared to non-users, which means that the former may eat more healthily. The interaction coefficient is positive and marginally significant, which indicates that starting to use NL increases dietary quality. The magnitude of the coefficient is 0.2192 and significant at the 10% level. The coefficient is somewhat robust depending on the model specification. In sum, using NL seems to improve dietary quality.

Looking at the regressions with energy share from fat, we see that NL switchers have a lower energy share of fat compared to the non-users. The difference is 1.2977 percentage points in the baseline model, but this decreases to 0.7312 percentage points once the full set of controls are added. When we look at the effect of using NL on fat intake, the results are ambiguous. The baseline model shows that starting to use NL is associated with an increase in the energy share from fat of 0.0635 percentage points, which is not significant. However, the coefficient becomes negative once the health controls are added, although it remains insignificant.

If we instead turn to the intake of added sugar, there is a negative, but non-significant interaction effect. The coefficient is −0.5044 in the baseline model and it remains somewhat robust to the model specification.

Lastly, looking at the intake of fiber, we see a positive interaction term that is independent of the model specification. The coefficient is 0.1155 and it is marginally significant in the baseline model. However, once all controls have been included, the coefficient decreases to 0.0964 and becomes insignificant. In conclusion, it seems that the improvement in dietary quality comes from a decrease in the intake of added sugar and an increase in the intake of fiber. Even though the changes in added sugar and fiber are insignificant, the combination of the two may drive the significant effect in the HEI.

#### 3.2.2. Description of Content

Figure 3 shows the average HEI and selected nutrient shares for the DoC switchers and non-users for examination of common trends. The regression results of the common trend are shown in Appendix A
Table A6. The DoC switchers have a relatively stable HEI in the observed period, whereas the non-users exhibit a decrease in dietary quality from 2009 to 2016. The regression analysis shows that the interactions between DoC and year are significant in 2009, 2010 and 2012. Hence, we should use the non-users as a counterfactual group with caution, since they exhibit a different dietary trend in the period prior to 2013.

The figure shows that the increase in energy share from fat slightly increases for both groups in the observed period. The test of common trends shows that we cannot reject common trends in the period 2009–2011. The interaction is marginally significant in 2012. The energy share from added sugar increases slightly for the switcher group from 2009 to 2011, but decreases slightly thereafter, whereas it increases slightly for the non-user group in the entire period. The regression analysis on common trends shows that none of the pre-treatment year interactions are significant, which means we cannot reject common trends. Lastly, the energy share from fiber remains relatively constant for the non-users in the observed period, while it decreases slightly for the switcher group from 2009 to 2010, increasing thereafter until 2012. The common trend regression shows that the year and DoC interactions are insignificant in 2009, 2011 and 2012. Hence, since there is only one year where there is a difference, I maintain the assumption of common trends.

The examination of common trends between the DoC switchers and the extended control group is shown in Appendix A
Figure A2 and Table A7. The conclusions are similar to those discussed above: the switchers and extended control group experience the same trend before 2013, except for dietary quality (HEI). As a robustness check, I checked another possible counterfactual group: the households that used DoC in both periods. When we test for common trends, the year and treatment interactions before 2013 are insignificant (see Appendix A
Table A10).

Table 6 shows the difference-in-difference regression results for the DoC. The first part of the table shows that there is a positive, albeit non-significant interaction effect, which is 0.0783 in the baseline model, although this decreases to 0.0594 once all the controls are added. The non-significant effect may be due to the opposite effect of some of the components in the HEI: The energy share from fat increases for the switchers relative to the non-users. This is robust across model specifications. The energy share from added sugar decreases, but this is not robust across model specifications and it remains non-significant. Lastly, using the DoC has a very small albeit negative effect on fiber intake. In sum, the evidence shows that using the DoC has no total effect on dietary quality.

Table 7 shows the HEI regression in which the counterfactual group is users in both years rather than non-users. In model (1) without controls, the interaction coefficient is 0.0708, which is close to the estimate when non-users are used as the counterfactual. However, when we add socio-economic controls, the difference-in-difference estimate of using the DoC becomes negative and close to 0. The estimate from the household fixed-effect model is larger than when non-users are used as the counterfactual group. None of the coefficients are significant. In sum, using a counterfactual group for which the assumption of parallel trends is more likely to hold, I still find that using the DoC has no effect on dietary quality.

#### 3.2.3. Multiple Label Use

Figure 4 shows the change in the HEI and selected nutrients from 2009 to 2016 for households that only start to use NL, households that only start to use the DoC, and households that start to use both of the labels in the observed period. Lastly, the non-users, who do not use either of the labels, are used as a control group.

The NL switchers, DoC switchers and non-users exhibit a decline in their dietary quality from 2009 to 2012, which indicates that the control is a good counterfactual to these two groups. Both switchers exhibit a somewhat constant dietary quality from 2009 to 2012. All of the three groups exhibit constant energy share derived from fat, added sugar and fiber from 2009 to 2012 with small fluctuations. For example, NL switchers exhibit a spike in their energy share from added sugar in 2010.

The regression of common trends shows that most pre-treatment years and treatment group interactions are insignificant and, therefore, I maintain the assumption of common trends between each of the label groups and the non-user group (see Appendix A
Table A8). The common trend assumption also holds for an extended control group (see Appendix A
Figure A3 and Table A9).

Table 8 presents the difference-in-difference regressions, where a third group is added—Both—which is the households that start to use both of the labels, as described in Section 2.2. The table show the same results as the two separate regressions: the effect of using DoC on the HEI is small and non-significant across the scenarios, while the effect of using NL is larger and marginally significant in some specifications. The coefficient on the interaction between year and Both is 0.1644 in the baseline model, which is larger than the coefficient on NL (0.1489) and DoC (0.0229). However, this is not robust across model specifications: when health and socio-economic controls are added, the coefficient on Both becomes smaller than the coefficients on NL and DoC. Furthermore, it should be noted that the households that start to use both labels have significantly higher dietary quality than non-users in 2013, as seen by the coefficient on Both.

Looking at the energy share from fat, the results from before change slightly: the effect of DoC is non-significant and non-robust. However, the effect of NL is negative, which is opposite to the conclusion when we only looked at NL. The effect of using both labels is a 1.3048 percentage point increase in the energy share from fat in the baseline model. The coefficient increases to 1.9596 and becomes significant once health and socio-economic controls are added. With respect to fat consumption, this indicates that using both labels can have a negative outcome with respect to health. Turning to added sugar, the results from before remain stable: using NL decreases the share of added sugar, while the results with respect to the use of DoC are ambiguous. Using both of the labels has a negative effect on the energy share from added sugar: the coefficient is −0.7103 in the baseline model and larger than the other label coefficients. Lastly, using both labels leads to ambiguous results with respect to fiber. The coefficient on both is 0.0368 in the baseline model, which is smaller than the coefficient on NL (0.1024), but larger than the DoC (−0.0190).

In sum, it seems that paying attention to more than one label has no overall effect on dietary quality as measured by the HEI, but there may be a positive effect on some nutrients, e.g., added sugar.

## 4. Discussion

The result with respect to the effect of NL is in line with the literature on FOP labels, which shows that information that is visible and easy to understand helps consumers to choose healthier products. However, it should be noted that the magnitude of the effect is relatively small, and only marginally significant. An increase of 0.2–0.3 corresponds to a 1–1.5% change compared to the baseline level of HEI in the switcher group. The effect on the energy share from sugar is a decrease of 0.4–0.5 percentage points, while the effect on the energy share from fiber is approximately a 0.1 percentage point increase. This is equivalent to the findings of Zhu et al. [32], which also find a decrease in sugar and increase in fiber intake in cereals after the introduction of FOP labels. Furthermore, they find a decrease in fat, where I find the opposite here. The zero effect of nutrition tables is equivalent with the finding from Patterson et al. [17]. Using both labels has no additional effect on overall dietary quality beyond using one label. These results suggest that consumers do not benefit from more information. There is, however, indications that the intake of some nutrients are affected (e.g., added sugar).

The results showed that approximately one in four main shoppers search for NL and one in three look at the DoC. This shows that some consumers pay attention, which is the first step for food labels to influence dietary quality. Though higher than other studies, there is still room for increasing that share. Summary statistics of consumers that searched for both labels show that they are more health conscious than those that do not attend to the label. This indicates that awareness of food labels could be raised among those that are less health conscious and show that there is a selection into food label use as already established in previous research. This point has partly been addressed by policy makers, as a campaign was initiated in 2017: the purpose of the campaign was to increase awareness of the Keyhole label for those without education in the age group 35+, which is a population group with low dietary quality [52].

It is not possible to show from this data whether the consumers are able to interpret the labels correctly and whether the misinterpretation was the reason that dietary quality did not increase much for those that started to use labels. A report on the Danish consumers’ understanding of the Keyhole label shows that consumers on average have reasonable understanding of the label, but there is still some confusion, e.g., some believe that the label is associated with ecology [44]. This could explain the small effect of NL on dietary quality if consumers associate the label with non-nutritional outcomes and suggest that one way to increase the influence of NL is to communicate their purpose more clearly to the consumers. The limited effect of the NL in this case could be due to the fact that the FOP NL group consists of the Keyhole/Whole Grain labels, which are binary in nature and might not be the most informative format compared to other FOP labels. For example, one laboratory study found that Traffic Lights are better at informing consumers about the healthfulness of a product compared to a binary label, which could explain why the FOP labels in this study had a limited effect [20]. The misinterpretation effect of FOP labels has been found in other studies of FOP labels [27].

The non-existent effect of the DoC may be because the label has been used for reasons unrelated to healthy eating, i.e., the DoC can be used to check ingredients rather than nutrients. However, the lack of an effect may also be due to a lack of understanding: nutritional tables, such as the DoC, require that consumers possess more nutritional knowledge compared to labels that are binary in nature [10]. However, experiments have shown that consumers do know how to read nutritional tables, but they require more time to process [24] and therefore lack of time could be a factor that explains the small effect of nutritional tables in real shopping data.

Furthermore, the performance of the labels depends on whether the producers display the label: as for the DoC, it is mandatory for all pre-packed food products. The Keyhole/Whole Grain label is, as previously mentioned, voluntary for producers to print. It is, therefore, important for label performance that eligible products carry the label. Unfortunately, there is no official register to examine whether all eligible products carry the label. Edenbrandt and Smed [53] find that 4%–13% of the products within selected product groups carry the Keyhole label. However, it is not known whether those that do not carry the label are eligible or not. If all eligible products do not carry the appropriate labelling, this could also explain the limited effect of FOP label use and this could be point of focus for the policy makers. However, as this study does not take the share of labelled products into consideration, this does not guarantee that the effect of labels will be influenced by mandating FOP labelling. Future research should examine this issue more carefully.

Lastly, the small and non-significant effect of the two labels in this study could be due to substitution between healthy and unhealthy food products. As a consumer learns about healthy alternative within one product group (e.g., by looking at Keyhole-labelled products), they could switch to more unhealthy alternatives in other product groups (thanks to the anonymous referee for making this point). These substitution patterns have also been found in a store experiment, where providing information on product healthiness increased healthy choices in some product groups, but had the reverse effect in other product groups [8]. Overall, the results suggest that binary FOP labels and BOP nutrition tables might not be the best tool to influence consumer behavior for all of the reasons mentioned above.

Even though the study design has the advantage that it is possible to follow individuals who begin to use food labels and thereby remove the effect of time-invariant characteristics and take time trend of a counterfactual group into account, it has limitations:

First, there is a selection into food label use, as previously discussed. It has been suggested that consumers who use food labels are more motivated to eat more healthily than others [39]. The selection is not fully taken into account in this study design. For this reason, the estimates show the average treatment effect on the treated rather than the average treatment effect for a random consumer. However, since each of the food labels had no, or only a small effect, it is likely that the conclusion will also apply to less-motivated households, since they are presumably less willing to switch to healthier options.

Second, the use of food labels is self-reported with respect to the household’s most recent shopping trip. If the use of food labels during the most recent shopping trip differs from general label use, it may represent a potential limitation to the study design. However, only 11% of the households in each year report that their most recent shopping trip was atypical, which makes it reasonable to assume that the most recent shopping trip is a good proxy for general behavior. Furthermore, studies of self-reported and actual label use show that there is a significant relationship between self-reported use and actual use or purchase of labelled products, which suggests that the self-reported use constitutes a good approximation of actual use [42,53].

Third, several studies point to the fact that nutritional knowledge is an important determinant of food label use [40,41,51]. However, nutritional knowledge was not used as a control variable, since it was not available in the data. Nevertheless, part of this was controlled for by including education, which has been shown to be highly correlated with nutritional knowledge [40,54,55].

Fourth, a small number of households may influence the significance of the results. Since a small share of the households in the panel pay attention to labels, an even smaller share change their label use behavior over time. This influences the number of households in the regressions and, thereby, the significance of the results, especially in the regression that includes both labels. The small number of households that changed label use behavior implied that it was not possible to analyze selected product groups, e.g., cereals, which have been widely used in food labels studies.

## 5. Conclusions

This paper investigated whether the use of FOP nutritional labels and BOP nutrition tables has an influence on dietary quality. This paper contributed by using panel data to investigate the effect of food labels on dietary quality as well as insights into whether there is an additional effect of using several labels. To establish causality between food label use and dietary quality, panel data was used to observe households that changed their label use behavior. These households were then compared to those that did not use labels as a counterfactual group. The results showed that the use of FOP nutritional labels increased dietary quality. However, the effect was small. The increase in dietary quality could be due to decreased intake of added sugar and an increased intake of fiber. Using the BOP nutrition tables had no effect on dietary quality. Taking the use of both labels into account shows that for some nutrients, using several labels to gather more nutritional information may result in a gain. Future research should examine this more carefully.

## Figures and Tables

**Figure 1 nutrients-12-01704-f001:**
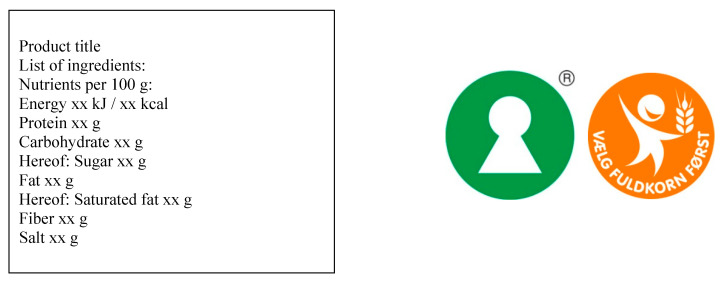
Example of Description of Content (DoC) (**left**) and nutritional labels (NL): the Keyhole and Whole Grain labels (**right**). Note: Pictures from [45,46].

**Figure 2 nutrients-12-01704-f002:**
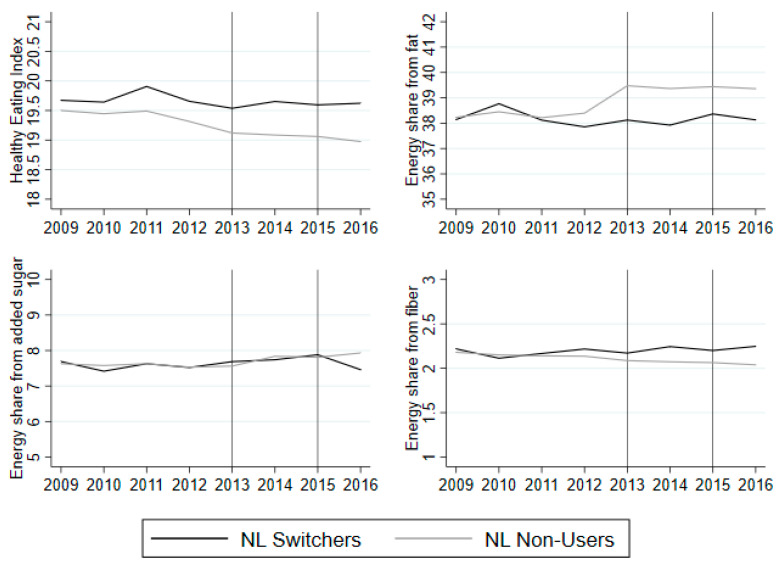
Average Healthy Eating Index (HEI) and nutrient shares from 2009 to 2016 for NL switchers (black) and non-users (grey). Note: The vertical lines indicate the years in which the questionnaires were sent to the panel. The average is based on households that answered the question on use of NL in 2013 and 2015. The averages are based on unbalanced panel from 2009 to 2016, as not all of the household stay in the panel from 2009 to 2016.

**Figure 3 nutrients-12-01704-f003:**
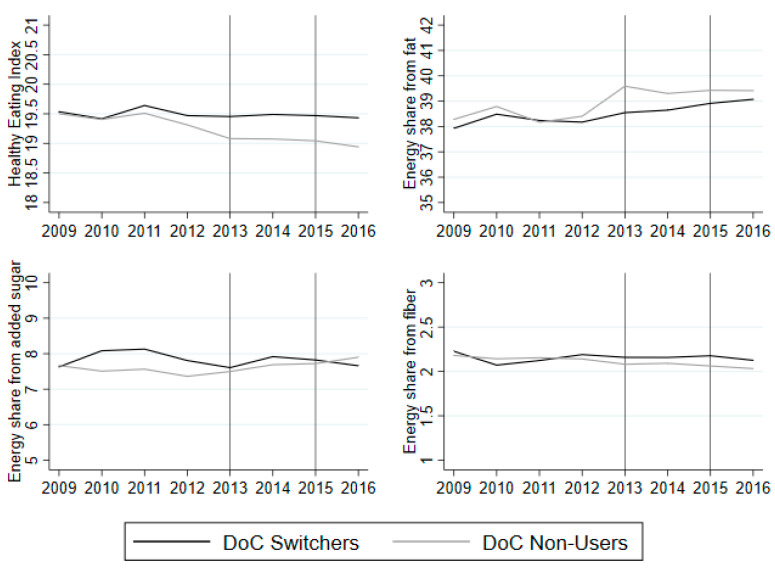
Average HEI and nutrient shares from 2009 to 2016 for DoC switchers (black) and non-users (grey). Note: The vertical lines indicate the years in which the questionnaires were sent to the panel. The average is based on households that answered the question on use of DoC in 2013 and 2015. The averages are based on an unbalanced panel from 2009 to 2016, as not all of the household stay in the panel from 2009 to 2016.

**Figure 4 nutrients-12-01704-f004:**
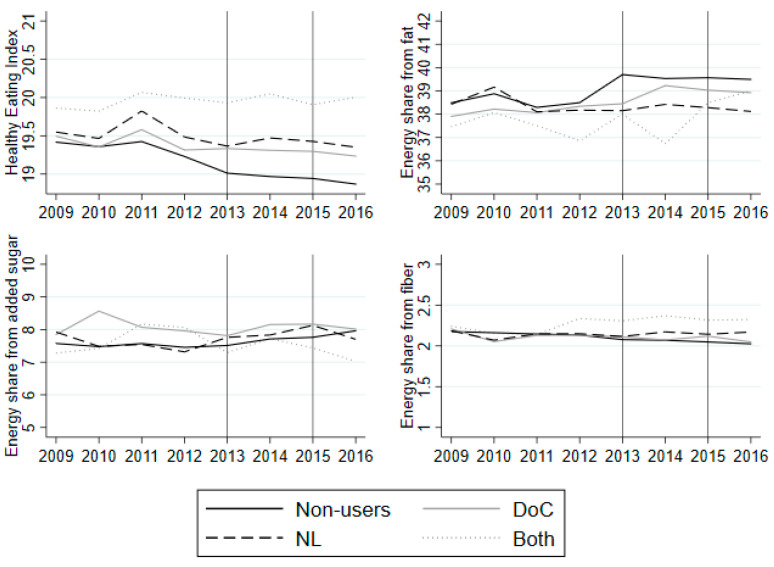
Average HEI and nutrient shares from 2009 to 2016 for NL switchers (dark grey dashed), DoC switchers (grey solid), Both switchers (light grey dotted), and non-users (black solid). Note: The vertical lines indicate the years in which the questionnaires were sent to the panel. The average is based on households that answered both questions on label use in 2013 and 2015. The averages are based on an unbalanced panel from 2009 to 2016, as not all of the household stay in the panel from 2009 to 2016.

**Table 1 nutrients-12-01704-t001:** Number of households according to NL or DoC use status in 2013 and 2015.

Group	NL	DoC
UN	149	161
UU	158	210
NU	131	153
NN	676	626
Total	1114	1150

Note: U refers to label users and N refers to non-label users in each of the years 2013 and 2015.

**Table 2 nutrients-12-01704-t002:** Number of households for each combination of label use in 2013 and 2015.

	NL				
DoC	UN	UU	NU	NN	Total
UN	56	30	6	57	149
UU	20	96	33	43	192
NU	7	15	40	87	149
NN	59	15	50	471	595
Total	142	156	129	658	1085

Note: U refers to label users and N refers to non-label users in each of the years 2013 and 2015.

**Table 3 nutrients-12-01704-t003:** Independent variables.

Variable Name	Type	Description
Socio-demographic variables		
Age	Continuous	Age of main shopper
Female	Dummy	Main shopper is a female
Household size	Continuous	Number of household members
Capital	Dummy	Whether household lives in the capital
Urban	Dummy	Whether household lives in an urban areas
Rural	Dummy	Whether household lives outside capital and urban areas (reference group)
No education	Dummy	Main shopper has no vocational or tertiary education (reference group)
Vocational education	Dummy	Main shopper has vocational education
Low education	Dummy	Main shopper has 1–3 years of tertiary education
High education	Dummy	Main shopper has 3 years or more of tertiary education
Income group 1	Dummy	Household income below 199,999 DKK (reference group)
Income group 2	Dummy	Household income in the 200,000–399,999 DKK interval
Income group 3	Dummy	Household income in the 400,000–599,999 DKK interval
Income group 4	Dummy	Household income above 600,000 DKK
Health variables		
Exercise hours	Continuous	Main shopper’s average weekly hours of exercise
Special diet	Dummy	Whether a household member eats according to a special diet
Healthy food attitude	Dummy	Main shopper agrees with statement “Healthy food is important to me”

**Table 4 nutrients-12-01704-t004:** Summary statistics of NL/DoC switchers and non-users in 2013.

	NL Switchers	NL Non-Users		DoC Switchers	DoC Non-Users	
	Mean	Mean	*p*-Value	Mean	Mean	*p*-Value
Age ^a^	55.98	56.81	0.532	57.96	56.06	0.100 *
Female ^a^	0.85	0.77	0.029 **	0.83	0.78	0.173
Household Size ^a^	1.90	1.98	0.448	1.80	1.99	0.015 **
Capital ^b^	0.08	0.12	0.002 ***	0.12	0.09	0.384
Urban ^b^	0.24	0.12		0.12	0.15	
Rural ^b^	0.69	0.76		0.76	0.76	
No Education ^b^	0.17	0.14	0.346	0.18	0.14	0.582
Vocational Education ^b^	0.39	0.39		0.40	0.39	
Low Education ^b^	0.12	0.18		0.16	0.17	
High Education ^b^	0.32	0.29		0.26	0.30	
Income Group 1 ^b,c^	0.17	0.16	0.064 *	0.20	0.16	0.310
Income Group 2 ^b, c^	0.49	0.38		0.43	0.40	
Income Group 3 ^b,c^	0.20	0.25		0.21	0.23	
Income Group 4 ^b,c^	0.13	0.21		0.16	0.21	
Exercise Hours ^a^	1.45	0.89	0.007 ***	1.16	0.97	0.301
Special Diet ^a^	0.28	0.29	0.836	0.32	0.30	0.691
Healthy Food is Important ^a^	0.86	0.66	<0.001 ***	0.82	0.70	0.001 ***
N	131	676		153	626	

Note: ^a^
*t*-test; ^b^ Chi-Square; ^c^ N is smaller for the Chi-Square test due to missing observations. The *p* shows the *p*-value of a test of means between the non-user and switcher groups. * *p* < 0.1, ** *p* < 0.05, and *** *p* < 0.01.

**Table 5 nutrients-12-01704-t005:** Difference-in-difference for NL.

**Dependent variable: HEI**					
	(1)	(2)	(3)	(4)	(5)
NL	0.4305 ***	0.2215	0.1997		0.1388
	(0.1572)	(0.1540)	(0.1575)		(0.1524)
Year	−0.1457 ***	−0.1437 ***	−0.1935 ***	−0.1547 ***	−0.1584 ***
	(0.0419)	(0.0462)	(0.0501)	(0.0437)	(0.0417)
NL*Year	0.2192 *	0.2885 **	0.2511 **	0.2172 *	0.2433 **
	(0.1187)	(0.1197)	(0.1250)	(0.1174)	(0.1229)
Observations	1614	1614	1503	1614	2085
R-Squared	0.016	0.083	0.138	0.019	0.125
**Dependent variable: energy share from fat**					
	(1)	(2)	(3)	(4)	(5)
NL	−1.2977 **	−0.8016	−0.7312		−0.4649
	(0.6370)	(0.6285)	(0.6428)		(0.6189)
Year	−0.0989	−0.1920	−0.4341 *	−0.0939	−0.3705 *
	(0.2202)	(0.2271)	(0.2441)	(0.2344)	(0.1979)
NL*Year	0.0635	−0.1099	0.0681	0.0591	−0.0222
	(0.5157)	(0.5330)	(0.5601)	(0.5141)	(0.5432)
Observations	1614	1614	1503	1614	2085
R-Squared	0.005	0.030	0.066	0.001	0.063
**Dependent variable: energy share from added sugar**					
	(1)	(2)	(3)	(4)	(5)
NL	0.0335	0.1811	0.0690		−0.0039
	(0.4061)	(0.4091)	(0.4292)		(0.4141)
Year	0.3374 **	0.3338 **	0.4514 ***	0.3692 **	0.3766 ***
	(0.1558)	(0.1581)	(0.1717)	(0.1618)	(0.1421)
NL*Year	−0.5044	−0.5701	−0.4755	−0.5071	−0.3918
	(0.3945)	(0.3987)	(0.4241)	(0.3937)	(0.4134)
Observations	1614	1614	1503	1614	2085
R-Squared	0.002	0.006	0.060	0.008	0.044
**Dependent variable: energy share from fiber**					
	(1)	(2)	(3)	(4)	(5)
NL	0.0944	0.0478	0.0236		−0.0015
	(0.0647)	(0.0638)	(0.0645)		(0.0631)
Year	−0.0480 **	−0.0497 **	−0.0577 ***	−0.0509 **	−0.0422 **
	(0.0196)	(0.0205)	(0.0221)	(0.0205)	(0.0185)
NL*Year	0.1155 *	0.1297 **	0.0964	0.1155 *	0.0945
	(0.0645)	(0.0652)	(0.0648)	(0.0638)	(0.0646)
Observations	1614	1614	1503	1614	2085
R-Squared	0.009	0.033	0.070	0.010	0.063
Health Controls	No	Yes	Yes	Yes	Yes
Socio-Demographic	No	No	Yes	No	Yes
FE	No	No	No	Yes	No
Extended Control Group	No	No	No	No	Yes

Note: Standard errors clustered at the household level. * *p* < 0.1, ** *p* < 0.05, and *** *p* < 0.01.

**Table 6 nutrients-12-01704-t006:** Difference-in-difference for DoC.

**Dependent variable: HEI**					
	(1)	(2)	(3)	(4)	(5)
DoC	0.3982 ***	0.2803 *	0.2670 *		0.2110
	(0.1515)	(0.1483)	(0.1520)		(0.1461)
Year	−0.1463 ***	−0.1116 **	−0.1573 ***	−0.1582 ***	−0.1422 ***
	(0.0444)	(0.0472)	(0.0528)	(0.0459)	(0.0417)
DoC*Year	0.0783	0.0840	0.0594	0.0656	0.0381
	(0.1050)	(0.1128)	(0.1251)	(0.1045)	(0.1197)
Observations	1558	1558	1448	1558	2148
R-Squared	0.012	0.079	0.132	0.024	0.121
**Dependent variable: energy share from fat**					
	(1)	(2)	(3)	(4)	(5)
DoC	−0.9995 *	−0.7379	−0.6878		−0.2986
	(0.5579)	(0.5547)	(0.5657)		(0.5447)
Year	−0.1497	−0.2830	−0.3792	−0.1307	−0.2897
	(0.2303)	(0.2353)	(0.2494)	(0.2459)	(0.1973)
DoC*Year	0.6468	0.5254	0.4276	0.6238	0.3741
	(0.5048)	(0.5179)	(0.5529)	(0.5008)	(0.5315)
Observations	1558	1558	1448	1558	2148
R-Squared	0.002	0.030	0.066	0.003	0.062
**Dependent variable: energy share from added sugar**					
	(1)	(2)	(3)	(4)	(5)
DoC	−0.0239	0.0416	0.0207		−0.1001
	(0.3842)	(0.3854)	(0.4095)		(0.3908)
Year	0.3653 **	0.3539 **	0.4087 **	0.3962 **	0.2804 **
	(0.1594)	(0.1597)	(0.1702)	(0.1640)	(0.1409)
DoC*Year	−0.2048	−0.1680	0.0299	−0.1676	0.1025
	(0.3364)	(0.3370)	(0.3569)	(0.3381)	(0.3436)
Observations	1558	1558	1448	1558	2148
R-Squared	0.002	0.005	0.051	0.010	0.040
**Dependent variable: energy share from fiber**					
	(1)	(2)	(3)	(4)	(5)
DoC	0.0984 *	0.0725	0.0588		0.0508
	(0.0597)	(0.0589)	(0.0582)		(0.0564)
Year	−0.0501 **	−0.0470 **	−0.0553 **	−0.0543 ***	−0.0259
	(0.0203)	(0.0210)	(0.0227)	(0.0208)	(0.0193)
DoC*Year	−0.0062	−0.0042	−0.0101	−0.0116	−0.0312
	(0.0450)	(0.0468)	(0.0480)	(0.0448)	(0.0463)
Observations	1558	1558	1448	1558	2148
R-Squared	0.005	0.028	0.063	0.014	0.062
Health Controls	No	Yes	Yes	Yes	Yes
Socio-Demographic	No	No	Yes	No	Yes
FE	No	No	No	Yes	No
Extended Control Group	No	No	No	No	Yes

Note: Standard errors clustered at the household level. * *p* < 0.1, ** *p* < 0.05, and *** *p* < 0.01.

**Table 7 nutrients-12-01704-t007:** Difference-in-difference with users in both years as counterfactual.

Dependent Variable: HEI				
	(1)	(2)	(3)	(4)
DoC	0.0429	0.0855	0.1298	
	(0.1708)	(0.1690)	(0.1710)	
Year	−0.1387 *	−0.1137	−0.0861	−0.1495 **
	(0.0757)	(0.0840)	(0.0871)	(0.0753)
DoC*Year	0.0708	0.0686	−0.0011	0.0636
	(0.1217)	(0.1300)	(0.1413)	(0.1213)
Observations	726	725	679	725
R-Squared	0.002	0.031	0.119	0.032
Health Controls	No	Yes	Yes	Yes
Socio-Demographic	No	No	Yes	No
FE	No	No	No	Yes
Extended Control Group	No	No	No	No

Note: Standard errors clustered at the household level. * *p* < 0.1, ** *p* < 0.05, and *** *p* < 0.01.

**Table 8 nutrients-12-01704-t008:** Difference-in-difference for NL, DoC and Both.

**Dependent variable: HEI**					
	(1)	(2)	(3)	(4)	(5)
DoC	0.3198 *	0.1867	0.1724		0.1353
	(0.1788)	(0.1743)	(0.1803)		(0.1761)
Year	−0.1476 ***	−0.1446 ***	−0.2080 ***	−0.1641 ***	−0.1800 ***
	(0.0495)	(0.0534)	(0.0583)	(0.0512)	(0.0513)
DoC*Year	0.0229	0.0269	0.0652	0.0181	0.0371
	(0.1363)	(0.1438)	(0.1562)	(0.1357)	(0.1528)
NL	0.3303 *	0.0911	0.0447		−0.0265
	(0.1931)	(0.1887)	(0.1903)		(0.1871)
NL*Year	0.1489	0.2682 *	0.2905 *	0.1531	0.2680 *
	(0.1517)	(0.1479)	(0.1506)	(0.1472)	(0.1483)
Both	0.9675 ***	0.7331 ***	0.7028 **		0.6487 **
	(0.2828)	(0.2739)	(0.2900)		(0.2877)
Both*Year	0.1644	0.1567	0.0045	0.1538	−0.0070
	(0.1580)	(0.1811)	(0.2090)	(0.1612)	(0.2059)
Observations	1392	1392	1291	1392	1577
R-Squared	0.028	0.098	0.156	0.023	0.138
**Dependent variable: energy share from fat**					
	(1)	(2)	(3)	(4)	(5)
DoC	−1.1024	−0.7771	−0.6632		−0.4377
	(0.6998)	(0.7001)	(0.7307)		(0.7092)
Year	−0.1584	−0.2327	−0.3392	−0.1249	−0.3038
	(0.2724)	(0.2777)	(0.2939)	(0.2870)	(0.2474)
DoC*Year	0.5172	0.4336	−0.0077	0.5187	−0.0116
	(0.6415)	(0.6574)	(0.7058)	(0.6384)	(0.6889)
NL	−1.3456	−0.7620	−0.5562		−0.1872
	(0.8516)	(0.8349)	(0.8297)		(0.8132)
NL*Year	−0.0224	−0.3075	−0.4360	−0.0334	−0.5121
	(0.6362)	(0.6577)	(0.6832)	(0.6272)	(0.6631)
Both	−1.7850 **	−1.2451	−1.3531 *		−1.0467
	(0.8284)	(0.8010)	(0.8007)		(0.7901)
Both*Year	1.3048	1.2371	1.9596 **	1.3198	1.8961 **
	(0.8560)	(0.8908)	(0.9397)	(0.8568)	(0.9312)
Observations	1392	1392	1291	1392	1577
R-Squared	0.007	0.034	0.070	0.006	0.068
**Dependent variable: energy share from added sugar**					
	(1)	(2)	(3)	(4)	(5)
DoC	0.0872	0.1844	0.0820		−0.1224
	(0.5076)	(0.5082)	(0.5461)		(0.5396)
Year	0.4068 **	0.3915 **	0.4448 **	0.4352 **	0.2568
	(0.1844)	(0.1859)	(0.1957)	(0.1893)	(0.1733)
DoC*Year	−0.0132	0.0188	0.2634	0.0189	0.4344
	(0.4484)	(0.4475)	(0.4701)	(0.4496)	(0.4651)
NL	0.1144	0.3205	0.1020		−0.1270
	(0.5573)	(0.5602)	(0.5893)		(0.5810)
NL*Year	−0.3762	−0.5105	−0.3600	−0.3867	−0.1293
	(0.5634)	(0.5719)	(0.5994)	(0.5644)	(0.5956)
Both	−0.2265	−0.0646	0.0384		−0.1390
	(0.5276)	(0.5325)	(0.5177)		(0.5013)
Both*Year	−0.7103 *	−0.6763	−0.5845	−0.6723	−0.4239
	(0.4199)	(0.4254)	(0.4607)	(0.4195)	(0.4481)
Observations	1392	1392	1291	1392	1577
R-Squared	0.003	0.009	0.064	0.010	0.057
**Dependent variable: energy share from fiber**					
	(1)	(2)	(3)	(4)	(5)
DoC	0.0386	0.0088	−0.0088		−0.0211
	(0.0672)	(0.0660)	(0.0653)		(0.0631)
Year	−0.0518 **	−0.0541 **	−0.0667 **	−0.0564 **	−0.0510 **
	(0.0241)	(0.0249)	(0.0271)	(0.0246)	(0.0227)
DoC*Year	−0.0190	−0.0194	0.0113	−0.0231	−0.0052
	(0.0575)	(0.0590)	(0.0609)	(0.0572)	(0.0591)
NL	0.0403	−0.0120	−0.0500		−0.0814
	(0.0776)	(0.0765)	(0.0770)		(0.0762)
NL*Year	0.1024	0.1275	0.1174	0.1010	0.1099
	(0.0847)	(0.0845)	(0.0838)	(0.0832)	(0.0832)
Both	0.2591 **	0.2052 *	0.1865		0.1629
	(0.1228)	(0.1223)	(0.1239)		(0.1250)
Both*Year	0.0368	0.0330	−0.0490	0.0318	−0.0551
	(0.0740)	(0.0819)	(0.0773)	(0.0744)	(0.0768)
Observations	1392	1392	1291	1392	1577
R-Squared	0.012	0.036	0.077	0.012	0.074
Health Controls	No	Yes	Yes	Yes	Yes
Socio-Demographic	No	No	Yes	No	Yes
FE	No	No	No	Yes	No
Extended Control Group	No	No	No	No	Yes

Note: Standard errors clustered at the household level. * *p* < 0.1, ** *p* < 0.05, and *** *p* < 0.01.

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
