# Peer review of "The Effect of Front-of-Pack Nutritional Labels and Back-of-Pack Tables on Dietary Quality"

_nutrients, 2020, doi:10.3390/nu12061704_

Round 1

Reviewer 1 Report

The authors report a well-conducted study on the use of nutrition labels in Denmark and their effect on the quality of the diet. My only concern is that the general result gets lost in the discussion, because the author describes many relatively minor-importance results extensively. The discussion section can thus be improved.

See the file enclosed for further comments.

Author Response

Thank you for the review. Please find my notes below (reference is to the line in original manuscript):

Line 112-119: Could you please elaborate on the comment? I am not entirely sure what format I should change?

Line 187: Thank you for pointing out that the term unusual needed supplementary descriptions. The respondents are asked to describe their latest shopping trip by the statement “The shopping trip was as it usually is”. They can either answer “Agree”, “Partly agree”, “Neither agree or disagree”, “Partly disagree”, “disagree”. Their shopping trip is defined as unusual if they disagree or partly disagree with the statement. In that sense it is not clear how is different from their usual shopping trip. This description has been added in a footnote for clarification

Line 202-204: Table 1 shows a grouping of households according to their label use behavior in the two years (2013 and 2015) and is meant as a descriptive table of a fixed grouping that is used in the subsequent analysis. I, therefore, do not believe it is relevant to test whether there is a significant difference in group sizes and what test would be relevant in this case. I test whether the groups differ in their characteristics in table 4

Line 367: The p-value does not appear for all tests because categorical groups (place of living, education, and income) are tested by a Chi-Square test (as indicated in table notes with the letters a and b) and thus there is only one p-value per group. The 0.000 is changed to <0.001

Line 419: Thank you for pointing out the inconsistencies. The tables and decimals in the text have now been changed so that they consistently display four decimals in the entire section.

Line 422: Even though the result is insignificant, I still believe that it is relevant to comment on the sign of the coefficient. The insignificant results could among other factors be due to a small sample of households that change label use behavior as noted in the discussion and therefore the sign is important for evaluation even though the result is insignificant

Line 424-425: The sentence has been deleted as is was too conclusive for a non-significant result.

Line 462: Thank you for pointing out the inconsistencies - The paper has been checked for abbreviations and the inconsistencies have been corrected.

Line 474: See comments for line 422

Line 488: Diff-in-diff has been corrected to difference-in-difference for consistency

Line 500+501: A 's' has been added to switchers as noted and 'the' has been deleted.

Line 516: The group name "Both" and the explanation of the group is described in detail in the method section. I have now added a reference to the section for the reader if they need clarification

Line 553: An 'a' is added as suggested.

Line 555: An 'a' is added as suggested

Line 563: Sentence is corrected as suggested

Line 615: 's' removed as suggested

Line 616: Thank you for pointing out the shortcomings of the section. This comment also made by the other reviewer. This section (line 614-619) has been deleted and a more specific discussion of the two labels investigated has been added further up in the paper to discuss the shortcomings of the labeling and what can be concluded from the data and other literature. 

The general point about the discussion: The discussion has been rewritten based on the note below as well as notes from another reviewer to focus more on the two labels in question.

Reviewer 2 Report

Please see the attached referee report

Author Response

Thank you for the review. Please find my comments below (reference lines is in the original manuscript):

Major points:

1 Thank you for noting that selection is important to address earlier as it plays a big role in the interpretation of the results. I have added this point to the method section also to high-light the shortcoming earlier in the paper 

2 Thank you for the important notes. I have removed the general conclusions from the discussion and focused the discussion on the labels that are investigated in the paper. The point about binary labels is really good and it has been added to the discussion. As the papers suggested in the comment do not consider binary labels I have looked into the experimental literature to find cases of studies that compare the performance of binary labels to other formats.

Unfortunately, the data do not provide a lot of information that could explain why the labels work as they do (or in this case do not work as intended), e.g. there is no information on whether the consumers can interpret them correctly and I, therefore, have to rely on a small number of studies and reports conducted in Denmark on the Keyhole label. As stated in the discussion approx 25 and 33 percent of the consumers search for nutritional labels and description of content respectively and there is thus scope to increase this share and focus could be made on those that are less concerned with health (Line 569-573) and one study found that consumers associate the Keyhole label with attributes not related to healthy eating (line 576-578). I added a section on label compliance of consumers, which could be important, but that this is hard to know the importance of in this study as well as your point about product substitution.

Minor points: 

1 This comment is very relevant The Whole Grain label is a part of the Nutritional Label group. Unfortunately it is not possible to break down the result into Whole Grain and Keyhole. The question of Nutritional Labels are asked in general (See line 169) and even though it is definitively relevant it is not possible to differentiate between nutritional labels in the analysis

2 Good idea. Legends have been added to all graphs

3 I have simplified table 1 and used the same terms as in table 2 as suggested.

4 The "and" has been deleted

5 I recognize the point and it is indeed discussable whether a 10 percent threshold is significant. However, as the sample is somewhat small, I believe that it is relevant in this case to use the 10 percent. I have added marginally significant in the text where results that are significant at the 10 percent level is discussed.